# Fractional Charge States in the Magneto-Photoluminescence Spectra of Single-Electron InP/GaInP_2_ Quantum Dots

**DOI:** 10.3390/nano11020493

**Published:** 2021-02-16

**Authors:** Alexander Mintairov, Dmitrii Lebedev, Alexei Vlasov, Andrey Bogdanov, Shahab Ramezanpour, Steven Blundell

**Affiliations:** 1Ioffe Physical-Technical Institute of the Russian Academy of Sciences, 194021 St. Petersburg, Russia; lebedev_84@mail.ru (D.L.); vlasov.scell@mail.ioffe.ru (A.V.); 2Electrical Engineering, University of Notre Dame, Notre Dame, IN 46556, USA; 3Department of Physics and Engineering, ITMO University, 197101 St. Petersburg, Russia; bogdan.taurus@gmail.com (A.B.); ramezanpourshahab@gmail.com (S.R.); 4SyMMES, IRIG, University Grenoble Alpes, CEA, CNRS, F-38000 Grenoble, France; blsteven1@club-internet.fr

**Keywords:** quantum dots, magneto-photoluminescence, Wigner localization, fractional quantum Hall effect, topological quantum computing

## Abstract

We used photoluminescence spectra of single electron quasi-two-dimensional InP/GaInP_2_ islands having Wigner-Seitz radius ~4 to measure the magnetic-field dispersion of the lowest *s*, *p*, and *d* single-particle states in the range 0–10 T. The measured dispersion revealed up to a nine-fold reduction of the cyclotron frequency, indicating the formation of nano-superconducting anyon or magneto-electron (*e*_m_) states, in which the corresponding number of magnetic-flux-quanta vortexes and fractional charge were self-generated. We observed a linear increase in the number of vortexes versus the island size, which corresponded to a critical vortex radius equal to the Bohr radius and closed-packed topological vortex arrangements. Our observation explains the microscopic mechanism of vortex attachment in composite fermion theory of the fractional quantum Hall effect, allows its description in terms of self-localization of *e*_m_s and represents progress towards the goal of engineering anyon properties for fault-tolerant topological quantum gates.

## 1. Introduction

The integer (IQHE) [1] and fractional (FQHE) [2] quantum Hall effects, which reveal a skipping-type dissipationless edge transport in two-dimensional (2D) electron (*e*) gas semiconductor structures in a strong perpendicular magnetic field, are two of the most important discoveries in the physics of the last decades of the 20th century. The IQHE provides a precise method of measuring the fundamental fine-structure constant and gave the first example of edge/surface topological states that are robust to external perturbations [3,4]. The FQHE yields very rich physics involving fractionally charged anyon states [5] and Majorana zero-energy modes [6]. These open up the possibility of building topologically protected quantum gates (TQG) [6,7], which promise to dramatically improve the operation of quantum computers and accelerate the creation of their compact hardware—one of the challenges of science and technology in the 21st century.

The IQHE is well understood in terms of noninteracting electrons and the quantization of their Hall conductance by integer Landau-level fillings, *ν* [8]. The FQHE—representing quantization by fractional fillings *ν* = *p/k* [9], where *k* and *p* are the rational integers—is instead considered to be a property of strongly interacting electrons condensing into incompressible liquids of anyon quasiparticles (QPs) having charge 1/*k*, as was suggested by Laughlin [10]. This liquid state is described by Laughlin’s wavefunction with *k* zeros (*k*-WZs) localized at the positions of each electron keeping them separated, thus reducing the interaction energy. The FQHE can also be described using the concept of composite fermion (CFs) particles consisting of an electron with 2*p* magnetic-flux-quanta (MFQ) attached, as was shown further by Jain [11]. In this approach, the FQHE states are the IQHE states of weakly interacting CFs and the CF wave-function of *k* Landau level state has 2*kp*-WZs, i.e., *ν* = *k*/(2*pk* + 1). Thus, both approaches are equivalent. While these theoretical approaches have described a rich set of experimental data since the discovery of the FQHE nearly 40 years ago, a microscopic description thereof remains elusive [12]. This results from the fact that they do not clarify the connection between WZs and MFQ, i.e., the microscopic mechanism of MFQ attachment, and its dependence on the localization of the electrons. This creates uncertainty in the interpretation of FQHE magneto-conductivity experiments related to the realization of topological quantum gates (TQG) [13,14,15].

Theoretical proposals [16,17] and a methodology [18] for topological quantum computing are based on Laughlin’s QP or Jain’s CF description of the FQHE, and recommend using a set of localized QPs/CFs pulled from the appropriate host liquids. Thus, experiments demonstrating the possibility of the creation of single localized fractionally charged QPs/CFs are a necessary step for the realization of TQGs, and are critical for our understanding of the microscopic nature of the FQHE. So far, however, the existence of localized QP/CF puddles/molecules has only been observed using high-spatial-resolution electrometry of 2D quantum Hall structures [19,20] or near-field photoluminescence measurements of quantum dots (QDs) [21], while the localization of single fractionally charged QP/CF has not been reported. 

Here, we demonstrate such localization and report the formation of fractionally charged states of single electrons, which gives insights into the mechanism of its coupling to MFQ. This clarifies the unresolved issues in the microscopic description of the FQHE and gives new perspectives for the realization of TQG. We use measurements of magneto-photoluminescence (magneto-PL) spectra of quasi-2D islands represented by self-organized semiconductor InP/GaInP_2_ QDs for the detection of the charge of single-electron states for a Wigner-Seitz radius *r*_s_~4 in a range of magnetic field 0–10 T. These QDs are large (~150 nm), providing a strong Wigner localization regime, together with in situ electron doping and a built-in magnetic field [22], and, thus, have unique properties compared to other QD material systems [23,24]. Using these, we measured a fractional charge in the range 1/3–1/9, which decreased linearly with an increase in the size of the QD and the angular momentum. This implies the formation of MFO vortexes and their fixed radius, which is limited by the Bohr radius. Thus, we revealed a critical role of localization in the coupling of the electron and MFO-vortexes, resulting in the formation of a magneto-electron (*e*_m_) particle. Its formation can be explained by the nano-superconducting properties of the electrons occupying quantized states in the Wigner localization regime and the self-generation of vortexes. The fractional charge of such *e*_m_ appears to be due to vortex screening of the electric field. Our observation shows that Laughlin’s *k*-WZs are related to close-packed topological vortex structures of *e*_m_, and provides an explanation for FQHE magneto-conductivity plateaus in terms of self-confinement of *e*_m_’s in a quasi-ordered potential modulation, which, in turn, opens up new possibilities of deterministic engineering of anyon particle properties for use in TQGs. 

## 2. Single-Electron InP/GaInP_2_ QDs

The measured InP/GaInP_2_ QDs (see Appendix A.1 for growth details) had a flat lens shape, a lateral size *D*~50–180 nm, up to 20 electrons and a Wigner-Seitz radius of up to *r*_s_~5, as described in [22,25]. The dots revealed a small elongation and asymmetry, which, in most cases, could be described as a combination of a ~5% elliptical distortion (*D*||/*D*⊥ = 1.05) and a 10% change of *R*⊥. The dots had up to ~0.2 content of Ga (*x*_Ga_) due to intermixing during growth, which resulted in an increase of the emission energy of the 00 transition *E*_0_ (see below) on ~150 meV. 

The InP QDs are surrounded by atomically ordered GaInP_2_ 20–50 nm size domains (AODs) having a CuPt_B_-type crystal structure, which is a [111]_B_-oriented GaP/InP monolayer superlattice [22]. Thus, they represent core-shell QD-AOD composites. Relaxation of the bonds of the domain was observed for some dots and resulted in an 80 meV decrease of *E*_0_. The domains generates a strong [111]_B_-oriented piezo-electric field, which produces electron doping of QDs and induces a built-in magnetic field (*B*_bi_) [21,26]. 

The structures studied had a relatively large dot density (~10^9^ cm^−2^) and we used an optical fiber probe cryo-magnetic near-field scanning optical microscope (NSOM) with a spatial resolution of 20-300 nm to measure the PL spectra of a single QD (see Appendix A.2 for details of NSOM technique). Since the QDs had a wide range of sizes, electron populations and *B*_bi_, the measurements presented certain technical challenges (including luck), the main one being the tedious scanning process to find the appropriate dot for a specific study, and separating its spectrum from neighboring dots. In our experiments, QDs with different electron populations were selected using analysis of the number of spectral features in PL spectra, as described in detail in [25]. The spectra were measured at 10 K and an external magnetic field *B*_e_ = 0–10 T.

In our NSOM measurements of InP/GaInP_2_ structures, we found nearly ten single-electron QDs from a set of ~200 dots. Here, we present the data for eight dots, denoted S00, S02, S03, S04, S09, S10, S11 and S12. Table 1 contains a list of these dots, their type (see below) and parameters, such as, quantum confinement (*ħω*_0_), *D*, *r*_s_, *E*_0_, *x*_Ga_, the separation of the electrons in a photo-excited state *d*_WM_, and *B*_bi_ (see [25]). The type and parameters were determined using magneto-PL spectra (see Appendix B.1 for spectra processing). 

From this set, the dot S00 was the only one having weak Wigner localization (WL) (*r*_s_~1) and the smallest size (60 nm), for which a singly charged exciton (trion) formed in the photo-excited state. The magneto-PL spectra of this dot are discussed in the Appendix A. The spectra were similar to those of the 2*e* excitonic (tetron) dot discussed in detail in [27], with a paramagnetic (negative) *B*-dispersion. 

For the rest of the dots, which had stronger Wigner localization (*r*_s_ > 2), the trion becomes unstable decomposing into a 2*e* Wigner molecule (WM) weakly bound to the photo-excited hole. Two of the dots, S02 and S09 had *r*_s_~2.5 (D < 100 nm), and were thus in the regime of the onset of WL; the rest had *r*_s_~4 (D > 100 nm) and were thus in the regime of strong WL. 

The change of *D* was from 85 nm (for S02) to 150 nm (for S04), which corresponded to a change in *ħω*_0_ from 2.5 to 0.6 meV. The *x*_Ga_ varied from 0.1 for S11 and S12 to 0.2 for S10 with *x*_Ga_~0.15 for the five remaining QDs. This corresponded to a change of emission energy *E*_0_ from ~1.7 to 1.8 eV. 

## 3. Magneto-PL Spectra: Experiment

Figure 1a compares the spectra of all seven WM QDs at *B*_e_ = 0 T plotted in Stokes shift energy units. As can be seen from Figure 1a, a zero-energy peak, denoted 00, and a few Stokes peaks, denoted 0*N*, which are the signature of 2*e*-WM emission [27], were observed for all dots. For dots S02, S09 and S10, only a single 01 Stokes component (SC) peak was clearly observed in the spectra, while *N* > 1 SCs had an order-of-magnitude smaller intensity. We denoted these single SC dots as S- type. For the rest QDs up to four SCs were observed; their intensities were larger or the same as that of 00 component. We denoted these multiple peak SC dots as S*-type. Note that the dots S11 and S03, which had the same *ħω*_0_, had nearly the same intensity distribution of 0*N* components, in which 02 and 03 components were dominant. They differed only by the emission energy, i.e., *x*_Ga_ (see Table 1).

Figure 1b presents magneto-PL spectra of the S11 and S12 QDs measured in the *B*_e_ range 0–10 T. It shows that the S11 and S12 QDs were observed at the same NSOM spectra, i.e., probe position, and thus, neighbored one another, being separated by~200 nm. Their spectra partially overlapped (at 1.696 eV for *B*_e_ = 0 T) and were separated using Lorentzian decomposition. At *B*_e_ = 0 T, the spectrum of the S11 dot also partially overlapped at 1.688 eV with the spectrum of another QD (D01m) with eight electrons and *ħω*_0_ = 3.5 meV. This dot will be discussed elsewhere. 

The intensity of the spectra of the S11/S12 dots versus *B*_e_ strongly increased/decreased, most probably due to the lateral shift of the NSOM probe toward the S11 dot. 

Thus, for *B*_e_ > 7 T, the spectral shape of S12 QD could hardly be analyzed, but the position of the center of gravity of the spectra could be clearly traced. 

When *B*_e_ increased, the S*-type peaks of the S11 dot merged into two at 4 T and then into a single peak at 6 T. We denote these types of spectra as D and P, respectively. A further increase in *B*_e_ led to a splitting of the P-type spectrum and its transformation into the S-type spectrum at *B*_e_ = 10 T. For the S12 dot, a P-type single peak shape seemed to appear at *B*_e_ = 4 T and 8 T, revealing some intermediary fine structure. For the QD S04 (see [27] and Figure 5a below), the P- and D-type spectra seemed to appear at 1 T and 4 T, respectively, but no S-type spectrum was observed over the entire range of *B*_e_ 0–10 T.

The key observation from the comparison of Figure 1a,b is that, in the spectrum of the QD S11 at *B*_e_ = 10 T and in the that of the QD10 at *B*_e_ = 0 T, the peak separation and their intensity distribution were nearly the same. The only difference was a slightly larger full-width-at-half-maximum (1.6 versus 1.2 meV), which may have been due to the larger value of *x*_Ga_ in S10 (see Table 1). These features, which indicate that these are nearly identical dots, are clearly seen in Figure 1c, in which we compare their spectra. In Figure 1d, we compare an NSOM image of the S10 dot (see [25]), *d*_WM_ estimated from the experimental spectrum (see below) and the calculated electron density for this dot for *B*_e_ = 0. The measured data and the calculation are in excellent agreement. This indicates that QD11 had zero internal magnetic field *B* at *B*_e_ = 10 T. This corresponded to *B*_bi_ directed opposite to *B*_e_ and equaled −10 T. Thus, we can conclude that the S*-type spectrum is a signature of *B*_bi_ and the fact that we did not observe S-type spectra for QD S04 corresponds to *B*_bi_ > 10 T. 

The total internal field in the QD is *B* = *B*_e_ + *B*_bi_, and a negative (positive) shift of the spectral components versus *B*_e_ corresponded to a positive (negative) shift versus *B*. This shift, outlined in Figure 1b by dashed lines connecting the peak maxima, revealed a weak positive *B*-dispersion in the S*-range and a negative *B*-dispersion in the S-, P- and D-ranges (see also the more detailed analysis in Figure 5a–c below). The average value of the dispersion of the P-spectra was a few times stronger than that of the S and S*, namely ~0.1 meV/T. For S04, a similar *B*-dispersion was observed [27]. The dispersion showed a dependence on the dot size, which may clearly be seen in Figure 1b by comparing the slope of the dashed lines of the P-type spectra of the S11 and S12 dots. The slope corresponds to a dispersion ~0.5 meV/T for S11 (*D* = 130 nm) and ~1 meV/T for S12 (*D* = 110 nm), thus increasing with a decrease in the dot size.

## 4. Magneto-PL Spectra: Phenomenology Description

### 4.1. Intensity Distribution of the Stokes Components 

In Figure 2a, we show a cartoon of the confinement potential and energy levels, together with a cross-section of the corresponding wave-functions *ψ_N_*(*r,θ*) and electron positions in the initial (IS) and photo-excited (PS) states of a single-electron QD having *r*_s_ = 3.8 (analog of S11). In the left and lower inserts, we show the calculated electron density distributions for the PS and the six lowest ISs (see Appendix A). Note that in Figure 2a and elsewhere, for PL transitions (see vertical arrows in the right), we use the notations 0*N*, where two numbers are radial quantum numbers (see Appendix B.2) of the PS and IS states, respectively, and, for single-electron states having the same angular momentum, the notations *N*’*s*, *N*’*p* and *N*’*d*, where *N*’ = *N* + 1, to number the states. It can be seen from Figure 2a that in the dot, a 2*e*-WM with *d*_WM_ = 70 nm formed by photon absorption generating an electron-hole (*e–h*) pair. An *e-h* recombination, resulting in photon emission corresponding to a transition between the lowest PS-IS 1*s*-states, which have total angular momentum *L_z_* = 0, left the residual 1s electron in a position shifted from that of the IS by a *d*_WM_/2. This shift broke the selection rule Δ*N* = 0 for an optical transition, and thus generated 0*N* (*N* > 0) PL SCs with an energy separation 2*ħω*_0_. The relative intensity of 0*N* PL transitions was equal to the ratio of the square of the overlap integrals $OI_{N}\left(d\right)=\int \psi_{0}\left(r+d,\theta \right)\psi_{N}\left(r,\theta \right)drd\theta$ for *d*= *d*_WM_/2. In Figure 2b,c we present numerically calculated ratios ${\left[OI_{N}\left(d\right)/OI_{0}\left(d\right)\right]}^{2}$ for *B* = 0 and *B*~7 T (see below) and corresponding PL spectra for *d* = *d*^WM^/2, respectively. These ratios are also shown as bars in Figure 1a. The comparison shows good agreement with the experiment. The enhanced intensity of higher *N* components in S*-dots was therefore due to *B*_bi_ and a corresponding increase of electron localization and *L*_z_ (see below).

For larger *L*_z_, *N*’*p_y_* and *N’d_y_* states appeared in the PL spectra. For these states, only zero Stokes energy components 1*p_y_* and 1*d_y_* were expected, owing to the spatial matching between their and the 1*s*^2^ PS maxima of the electron density distributions (see inserts in Figure 2a). The 1*p_x_* and 1*d_x_* states were expected to have nearly zero intensity, owing to the opposite phase of the upper and lower maxima, leading to a zero overlap integral. As we will show below, a single-peak P-type spectrum can be attributed to transitions involving the 1*d_y_* component of *L_z_* = 1 PS and a second peak in the D-type spectra to the 1*p_y_* component *L_z_* = 3 of PS. The S*-type spectra can be attributed to transitions involving the 1*s* component of the *L_z_* = 3 or 5 states.

### 4.2. Magic L_z_ Numbers of the Photo-Excited State

The increase of *L*_z_ in a few-electron 2D WMs versus *B* occurred via certain values known as “magic” numbers *L*_z_^MN^, which depend on the WM configuration [28]. *L_z_*^MN^ appeared because of the necessity of matching the spatial structure of the electron wave-functions and the spatial symmetry of the electron arrangement, and resulted from the competition between the Coulomb interaction and quantum and magnetic field confinement [29]. For 2e-WM, *L*_z_^MN^ are odd, i.e., *L*_z_ = 1, 3, and so on. They correspond to single electron PS configurations 1*s* + 1*p_x_*, 1*p_y_* + 1*d*_x_, 1*d_y_* + 1*f_x_*, … for *Lz* = 1, 1*p_x_* + 1*d_x_*, 1*s* + 1*f_x_*, … for *L_z_* = 3 and so on.

In Figure 3, we present the results of our calculations (see Appendix A for details) of the average orbital angular momentum 〈*L*_z_〉 = *L_z_* of the 2*e*-WM for an InP/GaInP_2_ QD having *D* = 150 nm (*r*_s_ = 4.3), corresponding to S04, versus *B* for *B* = 0-5 T. The inserts show the electron density distribution for *B* = 0, 1.0, 2.25, 3.0, 4.25 and 5 T. Results for smaller *r*_s_ are presented in Appendix A. 

Broadened plateaus at values of *L*_z_^MN^ =1 and 3 centered at *B*1~0.5 T and *B*3~1.25 T are seen in Figure 3. These plateaus were washed out for *L*_z_^MN^ = 7 and higher, after which *L_z_* showed a near-linear dependence on B. The plateaus shifted to higher *B* as *D* (*r*_s_) decreased, and for *D* = 110 nm (*r*_s_ =3.4), they were centered at fields that were two times larger. Assigning the P-, D- and S*-type spectra in Figure 1b to *L_z_* = 1, 3 and 5, the experimental *B_Lz_* fields (1.5, 4 and 8 T) seemed to be two times larger than the calculated ones (~0.7, 2 and 3 T). This may have indicated some mechanism for reducing the effective field acting on an electron, related to vortex formation (see below). 

The electron density distributions in Figure 3 show a strong decrease of the size of single-electron maxima *D_e_* versus *B*, which was nearly by a factor of two in the range from 0 to 5 T (~50 nm versus 30 nm). At the same time, *d*_WM_ did not change. Thus, the enhancement of higher-order SCs in the PL spectrum in S*-dots was related to a decrease in *D_e_*. Such a decrease can be formally described as an increase of the effective mass, since *D_e_*~ 1/*m**^2^. In Figure 2b, we present calculated *OI_N_*^2^ ratios using a nine-fold increase of *m** corresponding to *B*~7 T. The calculations yielded an S*-type order-of-magnitude increase of SCs having *N* > 1, confirming that these dots had strong *B*_bi_ (see also calculated spectra in Figure 2c).

### 4.3. Magnetic Field Shifts 

For the analysis of the *B*-dispersion of PL spectral components, we used the following expression for the emission energy of the ground state of a 2*e*-WM weakly bound to a hole (2*e*WM-*h*) in a parabolic potential [27]:E^2*eWM*-*h*^(B) = E_0_^2*eWM*-*h*^(0) + E_*C*,2*e*_(B) + E_*r*.*m*.,2e_(B) +*ħω*_0h_(B)(1)
where the first term is the “free” WM energy, the second term is the energy of the Coulomb interaction between the two electrons, the third term is the energy of the relative motion of the two electrons and the fourth term is the cyclotron energy of the hole (see Appendix B.2). In expression (1), we take advantage of the fact that cyclotron energy of center-of-mass c.m. motion does not depend on the number of electrons [30], i.e., *E*_c.m.,2*e*_(*B*) = *ħω*_0*e*_(*B*) + const. We also use the fact that *m_h_** > >*m_e_**, and we neglect the energy of the relative motion of electron and hole. We further assume, for simplicity, that the energy of the electron-hole interaction does not depend on *B*. The second term in (1) represents the difference between the energy of the 2*e* state of interacting and noninteracting electrons and it is approximately given by *E*_C,2*e*_(*B*) = 0.1$\frac{\omega_{c}}{\sqrt{m^{*}}}/\left(\sqrt[{4}]{\omega_{c}^{2}+\;4\omega_{0}^{2}}\right)$) (see [31]). The third term gives *L_z_*^MN^ transitions discussed above and makes much smaller contribution (see Appendix A). Thus, the *B*-dispersion of the 00 transition (i.e., 1*s* state) is
*E*_00_(*B*) = *E*^2*e*WM-*h*^(*B*)-*E*_0_^2*e*WM-*h*^(0)≈*ħω*_0*h*_(*B*),(2)
accounting for the fact that *ħω*_0_*h*(*B*) ≈ 2*E*_C,2*e*_(*B*). For *Ns*, *p_x,y_* and *d_x,y_* transitions, we find from the Fock-Darwin spectrum (see Appendix B.2)
*E*_0N_(*B*) ≈ *ħω*_0*h*_(*B*) − (2*N* + 1)*ħω*_0*e*_(*B*),*E_px,py_*(*B*) ≈ *ħω*0*_h_*(B) − 2*ħω*_0*e*_(*B*) ± 1/2*ħω*_c*e*_ (B),*E_dx_,_dy_*(*B*) ≈ *ħω*_0*h*_(*B*) − 3*ħω*_0*e*_(*B*) ± 1*ħω*_c*e*_(B),(3)

In Figure 4 we present a calculation of the energies given by Equation (3) for an electron charge *e** = 1 and 1/7, together with the experimental points for S* *N’s* peaks of S11 dot at *B* = 8 T, at which positive dispersion begins. The energies show a strong negative *B*-dispersion (~1 meV/T) for *e** = 1 resulting, from the small value of *m***_e_* and the fact that *m***_e_*<<*m***_h_*, i.e., a large *ω*_c*e*_ and *ω*_c*e*_ >>*ω*_c*h*_. Such large dispersion also have 1*p* and 1*d* states presented in Figure 4 too. For *e** = 1/7, a negative dispersion decreases down to −0.1 meV/T, owing to the decrease of *ω*_c*e*_ to values close to *ω*_c*h*_. 

The positive dispersion of the S*-region indicates *ω*_c*e*_ < *ω*_c*h*_ and for *e** = 1/7 results in a more than three-fold decrease of *m***_h_* (see example in Figure 4 for *B*> 8 T), which can indicate heavy-light hole mixing. A qualitative comparison of *B*-dispersions presented in Figure 4 with Figure 1b for *B* > 7 T shows that matching the peaks of the S- and S*-type spectra is achieved only for *e** = 1/7, thus indicating the observation of the fractional charge of single-electron states. The appearance of the S*-type spectra implies a recombination involving *e-h* states having *l_z_ = L_z_*^MN^ = 3 or 5, leaving the residual electron in the 1*s* state. Thus, positive dispersion is a signature for the mixing of the holes of the states having large angular momentum. 

## 5. Magneto-Electrons

### 5.1. Size-Dependence of Fractional Charge 

A quantitative comparison of *B*-dispersions and a fit using Equation (3) are shown for dots S04, S11 and S12 in Figure 5a–c, respectively. This gives fractional charge values 1/*k*, where *k* is the number of vortexes, versus *D* (see Table 2). In the figure, the PL spectra are shown as contour plots, on which peak positions (circles) and fitted curves were superimposed. Note that the fit includes *B*_bi_, which is -10 T for the S11 and S12 and −15 T for the S04 QDs. 

The fit makes it possible to consistently match all observed peaks to single-electron ISs *N’s*, 1*p_y_* and 1*d_y_* in these dots. We found that the fitting procedure was robust (in terms of self-consistency) against variations of level sequence, 1/*k* and *B*_bi_. 

The most straightforward fitting was obtained for S11 (see Figure 5b), which had a complete data set, in terms of the measured *B*-range, peak resolution and intensity. For this dot, the two low-field oscillations of the 1*s*-peak, which had an amplitude/period 0.3 meV/2T, could be attributed to switching between 1*s* and 1*p_x_*/1*p_y_* states; formation of P-type spectra could be attributed to the contribution of the 1*d_y_* state and its crossing with the 2*s* state at *B* = 3 T; the formation of the D-type spectra could be attributed to the contribution of the 1*p_y_* at *B* > 5 T; and the formation of S*-type spectra can be attributed to a positive dispersion at *B* > 7 T. 

Almost all these features were resolved in the S04 dot (see Figure 5a), accounting for the limited *B*-range due to the large *B*_bi_. For the S04 dot, however, the 2*s* state was not resolved, probably because of an overlap with the 1*d_x_* state, and S*-type peaks were smeared out/crossed because of the smaller energy separation. 

For the S12 dot, which had a limited set of identified peaks due to low PL intensity, it was possible to resolve (see Figure 5c) the following: the crossing between the 2*s* and 1*d_y_* states, manifesting the onset of P-type spectra at *B* = 3 T; the transition between D- and S*-type spectra at *B* = 6 T; and the positive *B*-dispersion of the 3*s* and 4*s* peaks in the S*-range. 

The measured fractional charge (see Table 2) was 1/9, 1/7 and 1/5 for the *N’s* states, 1/7 and 1/5 for 1*p* states and 1/5, ¼ and 1/3 for 1*d* states. This decreased with *D* and angular momentum. The number of vortexes versus *D* is shown in Figure 6 and reveals a linear dependence in the measured range. Also, for *D* = 150 nm, it shows a linear decrease versus *l_z_*. 

Note that the fit included a matching between *N’s* states in S- and S*-type spectra and revealed a suppression of the 1*s* (00) component in the S*-type spectra, which means that the peaks denoted 0*N* at *B*_e_ = 0 T in Figure 1b were actually 0(*N* + 1) peaks, as shown in Figure 5a,b. This corresponded to a much smaller “effective” *D_e_* (see lower plot in Figure 1b) due to vortexes and related WZs (see inserts in Figure 6 below).

### 5.2. Magneto-Electrons and Fractional Quantum Hall Effect 

Our results presented in Figure 5a–c and Figure 6 show self-generation of MFQ vortexes by a single electron in the strong Wigner-localization regime and imply a nano-superconducting characteristic of its quantum states. We denote such states as magneto-electron states or simply as magneto-electrons – *e_mk_*s. Note that for *l_z_* > 0, vortexes of opposite directions were generated for the positive and negative parts of wave-functions, which was thus related to Majorana zero energy modes. 

The *k*(*D*) dependence implies a critical size *D*_0_ for which fractional charge can be generated, corresponding to *k* = 2. This, in turn, implies hard-wall vortexes, i.e., the existence of a fixed vortex size *d*_v_ and close-packed spatial structures. Assuming conservation of linear *k(D*) dependence for *D* < 100 nm and an effective size of the electron density generating vortexes equal to *D*_v_ = *D*/2, we could estimate *d*_v_ ≈ *D*_0_/4 = 20 nm or a radius *r*_v_~10 nm for the *s*-state. This value was close to the Bohr radius *a*_B_* of InP/GaInP_2_ QDs (~8 nm see Appendix B.1 and Appendix A). Possible close-packed structures of *a*_B_*-size-vortexes for different *D* and single-electron states are shown in the inserts in Figure 6. The observed decrease of *k* versus *l_z_* may have been due to the reduction of *D*_v_ due the spatial “splitting” of electron density.

The vortexes of *e*_m_s represented current loops and could be considered as a “bound” charge part of the electron wave function. Thus, a fractional charge 1/*k* appeared, owing to the vortex screening the electric field. In InP/GaInP_2_ QDs, self-formation of *e*_m_s generating vortexes and *B*_bi_ may have been due to the dissipationless electron motion in a lateral electric field produced by the AOD GaInP_2_ shell. Thus, one can assume that for the S-type S10 dot, which had nearly the same D as the S*-type S11 and S03 dots, the lateral field was absent and *B*_bi_ was not generated. This may have been due to the different GaInP_2_-shell structure, in which AODs were weakly ordered because of large *x*_Ga_.

We should point out that different vortex-type single-electron eigenstates were considered theoretically in billiard models of square and related shape geometries in magnetic fields [32]. In this case, these states are excited states and vortex solutions arise because of the matching of the spatial geometry of the eigenstates to the shape of the confining potential. These vortex eigenstates appeared to be similar to the vortexes in mesoscopic superconductors of similar shape [33], which link these systems to *e*_m_ and give rise to the notion of the superconducting 2D “droplet” formed by a single-electron quantum state.

For *r*_s_~4, the FQHE regime begins at *B*~0.2 T, corresponding to *ν* = 1, where *ν* = 4*h*/(*π*D^2^)/*e*_0_/*B* is effective Landau level filling factor, assuming uniform charge distribution within QD. The formation of *e*_m*k*_ has thresholds B~0.2*k* T corresponding to *ν* = 1/*k* and *D_k_*~*ka*_B_*. This makes it possible to describe the FQHE, i.e., the appearance of corresponding conductance plateaus in quantum Hall (QH) bar structures, using a localization of *e*_m*k*_s in the disorder-induced potential fluctuations. The key effect in such a description is a quasi-ordering of the fluctuations [34], which is analogous to spontaneous symmetry breaking induced by quantum monitoring [35]. This results in a spatial modulation having a half-period equal to the spatial size *ka*_B_*. This implies a self-confinement of *e*_m*k*_s at the corresponding *ν*. Therefore, localized/delocalized *e*_m*k*_s exist in the range between 1/*k* and 1/(*k* +1). In this range, delocalized *e*_m*k*_s provide edge hopping conductivity. For IQHE, Wigner-crystal-type network patterns of the localized electron states, observed in a scanning electrometer probe experiment for *ν* = 1 [36], naturally emerge from such self-confinement.

Within this picture, the alternation of ¼ and ½ charge versus size of the resonator, i.e., the number of electrons, observed in Aharonov-Bohm phase interferometry measurements of quantum Hall (QH) bars for *ν* = 5/2 [14], can be explained by alternation of the size of *e*_m_ by the Majorana modes.

### 5.3. Topological Quantum Computing

While in our experiment, the effective density of electrons was ~10^10^ cm^−2^, the density used in QH bars for experiments aiming to realize TQGs [16] was nearly twenty times larger [14]. At such high density, the anyon *e*_m*k*_s are hard to control locally, making it difficult to measure their statistics [17] and perform elementary topological quantum computing operations, i.e., anyon interchange (braiding) and pairing (fusion) [16]. The same control seems more manageable at low densities; we have already demonstrated optically induced pairing and the interchange of *e*_m_s in the same InP/GaInP_2_ QDs, but with six *e*_m*k*_s [21]. Moreover, owing to *B*_bi_, these were done at zero magnetic field and at relatively high temperature (10 K), which was more practical. Also, in InP/GaInP_2_ QDs, the properties of *e*_m*k*_ anyons could be controlled by *Ne*, *D*, QD shape and *B*_bi_. For InP/GaInP_2_ QDs, TQG could be realized using adjacent nano-electrodes attached to single electron transistors [20]. Such a chip made it possible to move *e*_m*k*_s around each other and to control the local charge to measure the initialization and manipulation of q-bits. 

## 6. Conclusions

In conclusion, using measurements of the magneto-PL spectra of single-electron InP/GaInP_2_ quantum dots having Wigner-Seitz radius ~4, we demonstrated the existence of a magneto-electron state having a fractional charge. This is the anyon state, in which a few magnetic flux quanta vortexes are self-generated by a single electron in Wigner localization regime, providing nano-superconducting properties of quantum states. The emergence of the magneto-electron corresponds to the three- to nine-fold reduction of the cyclotron frequency of single electron Fork-Darwin states and its dependence from quantum dot size measured. These imply a close-packed arrangements of the magnetic-flux vortexes and make it possible to describe the fractional quantum Hall effect in terms of magneto-electron self-confinement in the disorder-induced potential fluctuations. Such self-confinement gives rise to the possibility of engineering of anyon properties (statistics) of magneto-electrons, and could be used to build anyon-based fault-tolerant topological quantum gates.

## Figures and Tables

**Figure 1 nanomaterials-11-00493-f001:**
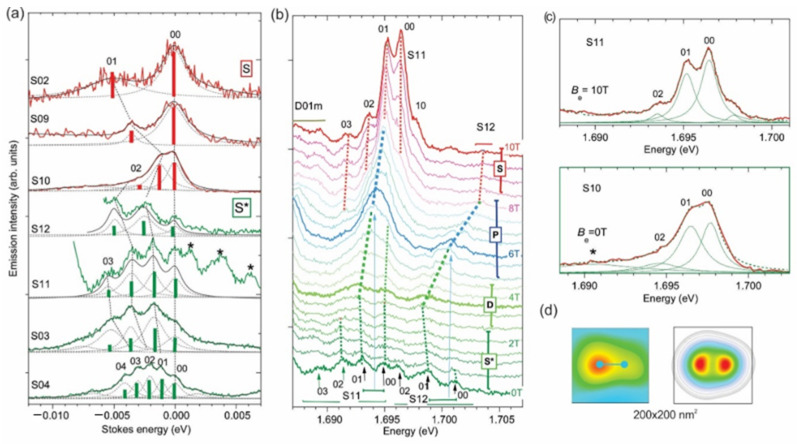
(**a**) Comparison of PL spectra (thick lines) S and S* 1e InP/GaInP_2_ QDs (see Table 1) at *B*_e_ = 0 T. Thin/dashed lines are Lorentzian contour decomposition of the spectra and bars are calculated 0*N* transition intensities (red and green for *B* = 0 and ~7 T, respectively, see text). The star symbols denote the neighboring dots. (**b**) PL spectra of S11and S12 InP/GaInP_2_ QDs measured at *B*_e_ = 0, 0.5, 1.0, … and 10 T. Dashed lines connect the positions of the peak maxima. (**c**) Comparison of PL spectra of S11 (**upper**) and S10 (**lower**). QDs measured at *B*_e_ = 10 and 0 T, respectively. Dashed lines are the spectrum and its constituents obtained from Lorentzian contour decomposition. The star symbols denote the neighboring dot. (**d**) Measured NSOM PL intensity distribution (**left**) and calculated 2e-WM electron density (**right**) for the S10 QD. The dumbbell in the left image corresponds to *d*_WM_, as-measured from the experimental PL spectrum.

**Figure 2 nanomaterials-11-00493-f002:**
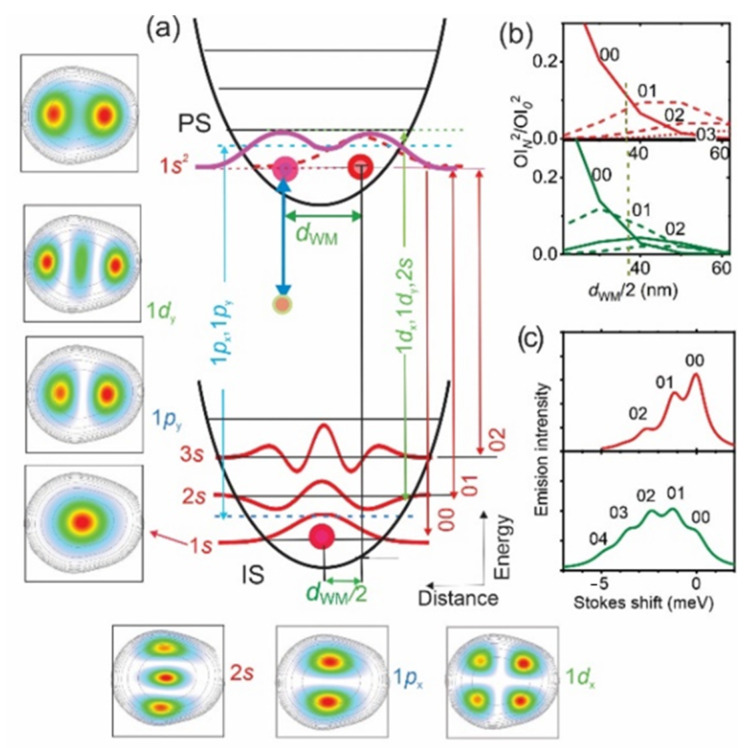
(**a**) Cross-sections of the parabolic confinement potential and energy levels of the initial (IS) and photo-excited (PS) state of a 1*e*-QD in the Wigner localization regime (*r*_s_ = 3.8). The curves are the cross-section of the wave-functions of the corresponding energy levels and vertical arrows show a set of PL transitions involving *Ns*, 1*p_x,y_* and 1*d_x,y_* states, the electron-density distribution of which are shown on the left and lower inserts (in the 200 × 200 nm^2^ frame). The thick arrow outlines a radiative recombination process and connects the photo-excited hole (smaller circle) with one of the electron (larger circles). (**b**) Square of the ratio of the overlap integrals *OI_N_* (see text) for *B* = 0 T (**upper**) and 7 T (**lower**). The dashed line shows the value of *d*_WM_/2 in (**a**). (**c**) PL spectra calculated using (**b**) for B = 0 T (**upper**) and 7 T (**lower**).

**Figure 3 nanomaterials-11-00493-f003:**
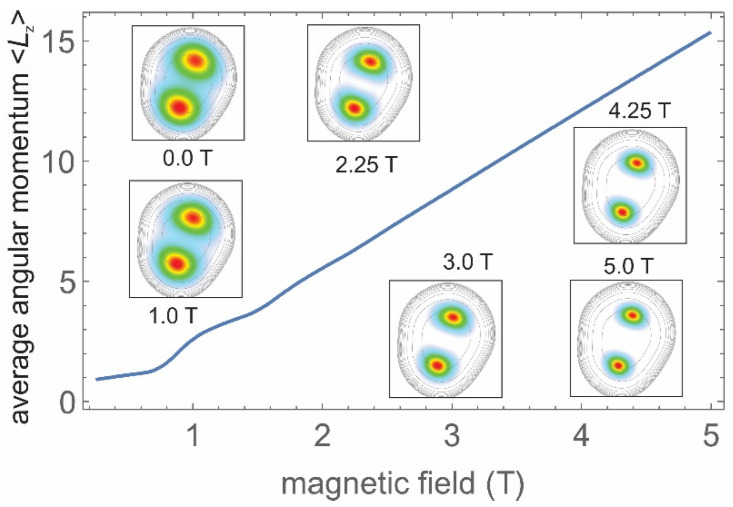
Average orbital angular momentum $\langle L_{z}\rangle$ calculated for the ground state of an InP/GaInP_2_ QD having *r*_s_ = 4.3. The inserts are densities for B=0 T, 1 T, and up to 5 T in the 200 × 200 nm^2^ frame. The contours indicate the deformed confining potential.

**Figure 4 nanomaterials-11-00493-f004:**
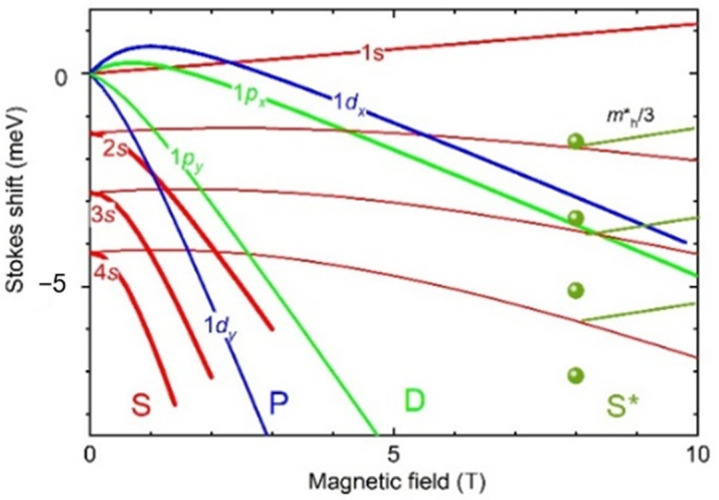
Stokes shift of *N’s*, 1*p_x,y_* and 1*d_x_,_y_* states versus magnetic field – thick(thin) curve for electron charge of 1(1/7), balls—experiment for S*. Transition to positive dispersion shown for S*-range corresponds to *m_h_**/3.

**Figure 5 nanomaterials-11-00493-f005:**
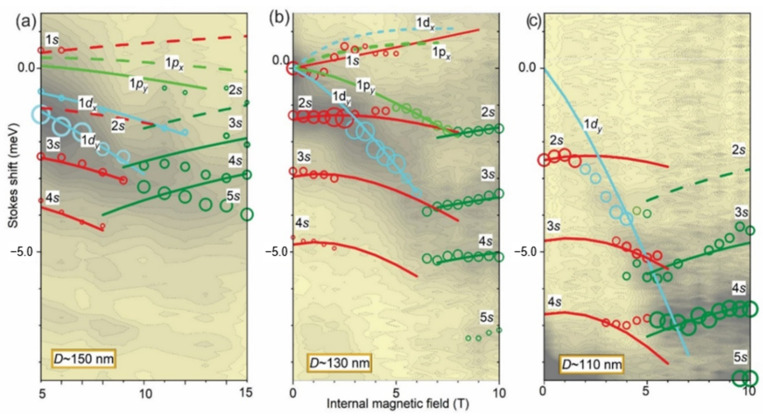
Contour plots of PL spectra of S04 (**a**), S11(**b**) and S12 QDs (**c**), peak positions (circles) and their Fock-Darwin spectra fit using Equation (3) versus internal magnetic field *B* = *B*_e_ + *B*_bi_.

**Figure 6 nanomaterials-11-00493-f006:**
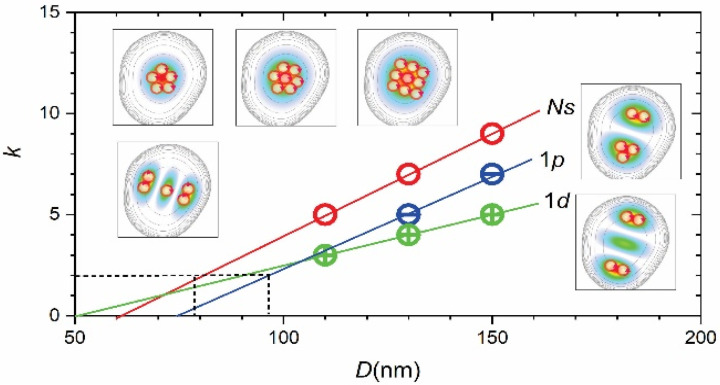
Number of vortexes (circles) versus dot size. Inserts show electron densities and vortex arrangement for 1*s* (upper, *D* = 110, 130 and 150 nm), 2s (lower left, *D* = 150 nm), 1*p* and 1*d* (right, *D* = 130 nm). Solid lines outline linear approximation of *k*(*D*), dashed lines show *k* = 2 (horizontal) and critical *D* for vortex formation (vertical).

**Table 1 nanomaterials-11-00493-t001:** Parameters of electronic states of single electron InP/GaInP_2_ QDs measured in magneto- PL spectra.

No	Type	*ħω*_0_ (me)	*D*^(a)^ (nm)	*r* _s_	*E*_0_ (eV)	*x* _Ga_	*d*_WM_ (nm)	*B*_bi_ T
S00	trion	4.0	60	~1	1.720	0.15	-	0
S02 [25]	S	2.5	85	2.5	1.724	0.15	35	0
S09 [25]	S	2.0	90	2.6	1.712	0.15	40	0
S10 [25]	S	0.7	140	4.1	1.798	0.2	75	0
S12	S*	1.25	110	3.4	1.705	0.1	65	10
S03 [25]	S*	0.8	130	3.8	1.765	0.15	70	10
S11	S*	0.8	130	3.8	1.695	0.1	70	10
S04 [27]	S*	0.6	150	4.3	1.743	0.15	80	15

^(a)^ Averaged value (between *D*_||_ and *D*_⊥_) of the size of the area containing 96% of calculated electron density.

**Table 2 nanomaterials-11-00493-t002:** Measured fractional charge of single electron states.

Quantum Dot	Dot Size *D* (nm)	Fractional Charge 1/*k*		
		*N’s*	1*p*	1*d*
S04	110	1/9	1/7	1/5
S11	130	1/7	1/5	1/4
S12	150	1/5	-	1/3

## Data Availability

The data that supports the findings of this study are available within the article and its Appendix A.

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
