# Peer review of "Fractional Charge States in the Magneto-Photoluminescence Spectra of Single-Electron InP/GaInP2 Quantum Dots"

_nanomaterials, 2021, doi:10.3390/nano11020493_

Round 1

Reviewer 1 Report

In this work, the authors revealed the magnetic-field dispersion of the lowest s, p and d single-particle states of single-electron InP/GaInP2 quantum dots in the range 0-10 T through the magneto- PL spectra. It is an interesting and novel phenomenon in the formation of nano-superconducting anyon or magneto-electron (em) states. The experimental phenomena about this work are abundant and the explanations are sufficient. The novelty of this work is highlighted, which will certainly to attract many readerships.  It will be suitable for publication in this journal after the authors address the following points:

  1. The author should give the detail characterization of the sample.
  2. The main text shows the magneto- PL spectra merely, however, the optical properties in nanocrystals under the external field is a very complicated issue. Such as the bright−dark state relaxation. The authors should perform the time-resolve PL(TRPL) to obtain conclusive evidence.
  3. The temperature dependent magneto- PL spectroscopy studies may help the authors understand more details.
  4.  Can it be compared with previous research results?

Reviewer 2 Report

The manuscript touches a new an extremely interesting topic concerning the appearance of fractional charge states in the emission spectra from individual  quantum dots. The observation of anyons with various fractional charges by methods of optical spectroscopy would be a great breakthrough opening a new field of research. My major concern is, however, the poor quality experimental data. Claiming this important result would require more clear experimental data. Instead, the quality of the emission spectra presented in  Figures 1 and 5 is not good enough, in my opinion. First of all, there are emission spectra from several dots overlapping with each other. Secondly, the position and shape of each spectral line cannot be clearly followed in an external magnetic field. Concluding, I recommend the rejection of the manuscri pt due to  overinterpretation of the experimental results.

Reviewer 3 Report

The authors prepared and investigated InP/GaInP2 quantum dots from the viewpoint of their magnetic-field variable photoluminescent spectra which were gathered for the selected single-electron islands on the nanomaterial. These data were thoroughly analysed to demonstrate the existence of a magneto-electron state possessing fractional charges, which is related to the fractional quantum Hall effect. This work is a complete piece of science contributing to the broad research on quantum Hall effects generated in the QD semiconductors under the application of a magnetic field which was nicely presented by the authors in the introduction. Therefore, I recommend the publication of this manuscript after a minor revision related to the following points:

1. The introduction is focused on the objective physical effects. However, a word could be added about the field of QDs semiconductors, their tunable luminescent properties, and about the selected material of InP/GaInP2.

2. The variation in the xGa within the series of investigated QDs should be commented.

3. The authors stated that a set of ca. 200 dots were checked to successfully gather the satisfactory luminescent data for 8 dots. It will be useful for the reader to know some details about the other examined dots. Is it the main problem related to the overlapping othe emission bands from the neighbouring dots? Some details how the reported QDs were selected and the other were excluded will be valuable.

4. The insets in Figure 3 are extremely small. It should be corrected.

5. In the structure growth section, the authors briefly described the procedure to prepare the sample. Is it the modified published procedure? If yes, the reference should be indicated.

6. The conclusion section should be slightly expanded towards the comment on the applied methodology including the pathway from the experiment to the final conclusion on the existence of a magneto-electron state. It is also not clear how the described findings contribute to the prospective construction of fault-tolerant topological quantum gate. A sentence of explanation will be valuable.

Round 2

Reviewer 2 Report

In order to be more convincing concerning the quality of the measurements, the authors should give a statement about the stability of the system. Do the measuremets give exactly the same results when repeating the scans in an external magnetic field? 

The evidence of fractional charge states  in quantum dots would be an important result of a great interest for the community. That is why I'm so worried about the reliability of the measurements.

Author Response

File with the answer is attached

Round 3

Reviewer 2 Report

I cannot agree with the statement from the last response, that the stability of the system "is not critical issue in our measurements and conclusions." When having an unstable setup and measuring several PL-lines  simultaneously, one can find any spectral position dependence fitting to any theory after performing sufficiently large number of scans. 

On the other hand, the data shown in the last response shows a reasonable degree of repeatability concerning the spectral position dependence on the magnetic field. Therefore, I have changed my opinion and recommend the acceptance of the manuscript. Further judgment about the reliability of these results I will let to the Readers. 

Author Response

We are glad for a recommendation of the Referee to accept our manuscript